# Exposure to polychlorinated compounds and cryptorchidism; A nested case-control study

Jonatan Axelsson [1,2,3] *, Kristin Scott[2], Joakim Dillner[4], Christian H. Lindh[2], He Zhang [3], Lars Rylander[2], Anna Rignell-Hydbom[2]

**1** Reproductive Medicine Centre, Skåne University Hospital, Malmö, Sweden, **2** Division of Occupational and Environmental Medicine, Department of Laboratory Medicine, Lund University, Lund, Sweden, **3** Reproductive Medicine, Department of Translational Medicine, Lund University, Malmö, Sweden, **4** Department of Medical Epidemiology and Biostatistics, Karolinska Institute, Stockholm, Sweden

* jonatan.axelsson@med.lu.se

**Data Availability Statement:** Due to the personal data collected and potentially identifying information contained within the data, data are available upon request. Ethical approval by The

## Abstract

### Background

Maldescended testes or cryptorchidism is a genital birth defect that affects 2–9% of all male new-borns. Over the last 40 years there have been reports of increased prevalence in countries like the US, the UK and the Scandinavian countries. This possible increase has in some studies been linked to a foetal exposure to chemical pollutants. In this matched case-control study, we analysed maternal serum samples in early pregnancy for three different organochlorine compounds, to investigate whether the levels were associated with the risk of cryptorchidism.

### Method

Maternal serum samples taken during the first trimester of pregnancy from 165 cases (boys born with cryptorchidism) and 165 controls, matched for birth year and maternal age, parity and smoking habits during the pregnancy, were retrieved from the Southern Sweden Maternity Biobank. The samples were analysed for 2,2',4,4',5,5'-hexachlorobiphenyl (PCB-153), dichlorodiphenyltrichloroethane (*p,p'*-DDE) and hexachlorobenzene (HCB), using gas chromatography mass spectrometry. Associations between exposure and cryptorchidism were evaluated by conditional logistic regression.

### Results

We found no statistically significantly associations between exposure to these compounds and cryptorchidism, either when the exposure variables were used as a continuous variable, or when the exposure levels were divided in quartiles.

### Conclusion

We found no evidence of an association between maternal levels of PCB-153, *p,p'*-DDE or HCB during the pregnancy and the risk of having cryptorchidism in the sons.

Swedish Ethical Review Authority may be necessary, and requests for data may be sent to LUPOP - Lund University Population Research Platform to the email address: lupop@ed.lu.se.

**Funding:** ARH: 2008-32038-63634-60, The Swedish Research Council, https://www.vr.se/english.html The funders had no role in study design, data collection and analysis, decision to publish, or preparation of the manuscript.

**Competing interests:** The authors have declared that no competing interests exist.

## Introduction

Maldescended testis, or cryptorchidism, is a genital birth defect that affects up to 8% of newborn boys [1], which is associated with a risk of testicular cancer and impaired fertility later in life [2]. An increased prevalence over the last 40 years has been reported in some countries, such as in the US [3], the UK [4] and the Scandinavian countries [5]. The knowledge about risk factors for cryptorchidism is still scarce [6], but due to a possible increase in incidence, exposure to environmental chemicals, especially so called endocrine disruptors, has been suggested as a potential risk factor [7]. Environmental chemicals include several lipophilic organochlorine compounds, often called persistent organochlorine pollutants (POP), such as polychlorinated biphenyls (PCB), dioxins and pesticides like dichlorodiphenyl trichloroethane (DDT) and hexachlorobenzene (HCB) [8]. These chemicals are resistant to degradation and accumulate in food chains [9]. Despite the fact that the majority of POPs have been restricted or banned in most countries during the 1970´s and 1980´s, many are still found in humans due to their persistence [10]. POPs are known to cross the placenta and serve as a prenatal source of exposure for the developing foetus [11, 12]. Although recent reviews have found little evidence for associations between potential endocrine-disrupting chemicals, including the mentioned POPs, and cryptorchidism [6, 13, 14] associations have been suggested to be stronger in certain contexts [6], such as an association between DDE and cryptorchidism, hypospadias and testicular cancer, taken together as a single outcome representing male reproductive disorder [14].

In this study we used first trimester maternal serum samples from a large population-based maternity biobank in Southern Sweden, to establish POP exposure in utero, similar to a previous study [15]. The samples were analysed for 2,2',4,4',5,5'-hexachlorobiphenyl (PCB-153) as a biomarker for PCB exposure, since PCB-153 has been found to correlate very well with total PCB concentration in plasma and serum from Swedish individuals [16–18]. The samples were also analysed for 1,1-dichloro-2,2-bis (*p*-chlorophenyl)-ethylene (*p,p*'-DDE), which is a persistent metabolite of DDT, and for HCB. Maternal serum levels of all three chemicals have been found to correlate well with foetal levels [19].

## Material and methods

The study was performed in accordance with the Declaration of Helsinki. The study was approved the local Ethical Review Board at Lund University with the approval number 177/2005. The ethical approval included that a specific consent for the study was not needed.

### Study population

The Southern Sweden Microbiology Biobank (SSMB) is a population-based health care biobank containing serum samples submitted primarily for virus diagnostics. All samples have been stored during 1986 and from 1988 onwards, resulting in about 1.3 million serum samples from 524.000 donors at the time when the linkage for the current study was done in 2002. A separate cohort in the SSMB is the Maternity Biobank, containing serum samples taken during the first trimester of pregnancy from women undergoing routine screening for HIV, hepatitis B, syphilis and rubella immunity at the maternity care, a screening in which virtually all pregnant women participate.

Cases (boys born with cryptorchidism), were identified by linkage between the SSMB, the Medical Birth Registry (MBR), the Malformation Registry and the In-patient Registry, and resulted in 640 boys with cryptorchidism. For 186 (29%) of these identified cases, we could localize a sample in the biobank that contained sufficient amount of serum to be included in the study. Any additional information about the pregnancy was retrieved from the MBR. For

165 of these boys, we were able to find at least one control boy within the biobank—matched on birth year and maternal age, parity and smoking habits early in the pregnancy. For four of the cases, two control boys could be found, ending up with 169 control boys. For those four cases with more than one control, we randomly included one of the two controls, ending up with 165 control boys. Boys with hypospadias or any major malformation were excluded.

### Determination of PCB-153, p,p´-DDE and HCB

PCB-153, *p,p'*-DDE and HCB were extracted from 0.005 ml aliquots of serum by solid phase extraction (Strata SDB-L 200 mg; Phenomenex, Torrance, CA, USA) using on-column degradation of the lipids and analysis by gas chromatography mass spectrometry by the use of negative-ion chemical ionization. Selected ion monitoring were performed at m/z 360, 318, 284 for PCB-153, *p,p'*-DDE and HCB, respectively, and $^{13}C_{12}$-labeled PCB-153, $^{13}C_{12}$-labeled *p,p'*-DDE and $^{13}C_{12}$-labeled HCB were used as internal standards at m/z 372, 330 and 290, respectively. The relative standard deviations, calculated from 100 samples analysed in duplicate at different days, was 5% at 2 ng/ml for PCB-153, 8% at 7 ng/ml for *p,p'*-DDE and 6% at 0.3 ng/ml for HCB. The limit of detections were set at 0.05 ng/ml, 0.1 ng/ml and 0.02 ng/ml for PCB-153, *p,p'*-DDE and HCB respectively. The analyses of PCB-153, *p,p'*-DDE and HCB were part of the Round Robin inter-comparison program (Professor Dr med. Hans Drexler, Institute and out-patient Clinic for Occupational, Social and Environmental Medicine, University of Erlangen-Nuremberg, Germany) with analysis results within the tolerance limits. Levels of PCB-153 and DDE were missing in four controls and two cases.

### Statistical analyses

The association between the levels of maternal POP concentrations as continuous variables and the risk for cryptorchidism in the offspring was evaluated by conditional logistic regression in SPSS: https://www.ibm.com/support/pages/11-matched-case-control-studies-conditional-logistic-regression. We were given the odds ratios (OR) of being a case for one unit increase in level of exposure [20] as the risk measure with 95% confidence intervals (95% CI). After analyses with POP concentrations as continuous variables, the concentrations (PCB-153, *p,p´*-DDE and HCB) were categorized into four equally sized groups (quartiles) based on the distributions among the controls, after which the same statistical analyses as above were performed, assessing the odds ratio of being a case for one unit increase in exposure quartile. Due to the relatively high correlation between the concentrations of PCB-153, *p, p´*-DDE and HCB concentrations (r varied between 0.37 and 0.52), we did not include these exposure measures simultaneously in the models in our original analyses.

Finally, as a first sensitivity analysis, we repeated the analyses above after only including boys that were born full term (at least 37 weeks of pregnancy length), which was the case in 320 boys. As a second sensitivity analysis we instead included all POP levels as continuous variables in the model at the same time, in all the boys.

## Results

Variables matched for in the cases and controls can be found in Table 1.

**Table 1. Statistics of variables matched for in the studies boys and their mothers.**

| | Median birth year | Mean maternal age (years) | Mean maternal parity | Proportion of mothers smoking |
|---|---|---|---|---|
| Cases | 1995 | 29 | 1.6 | 9.6% |
| Controls | 1995 | 28 | 1.6 | 10% |

**Table 2. Maternal serum concentrations of (and quartiles of exposure to) PCB-153 (n = 324), *p,p'*-DDE (n = 324) and HCB (n = 330) during pregnancy in cases of cryptorchidism and matched controls, as shown in all boys (n = 330) and those only born full term (n = 320).**

| Compound | Quartile levels | All boys | | Full term only | |
|---|---|---|---|---|---|
| | | Cases | Controls | Cases | Controls |
| PCB-153 (median values) | | 0.45 ng/mL | 0.47 ng/mL | 0.45 ng/mL | 0.48 ng/mL |
| | | Number of boys | Number of boys | Number of boys | Number of boys |
| PCB-153 quartile 1 | <0.23 ng/ml | 40 | 40 | 40 | 39 |
| PCB-153 quartile 2 | 0.24–0.46 ng/ml | 43 | 39 | 43 | 33 |
| PCB-153 quartile 3 | 0.46–0.69 ng/ml | 36 | 42 | 36 | 42 |
| PCB-153 quartile 4 | 0.70> ng/ml | 44 | 40 | 44 | 37 |
| *p,p'*-DDE (median values) | | 1.1 ng/mL | 1.1 ng/mL | 1.07 ng/mL | 1.05 ng/mL |
| | | Number of boys | Number of boys | Number of boys | Number of boys |
| *p,p'*-DDE quartile 1 | ≤0.05 ng/ml | 35 | 44 | 35 | 41 |
| *p,p'*-DDE quartile 2 | 0.12–1.0 ng/ml | 45 | 35 | 45 | 33 |
| *p,p'*-DDE quartile 3 | 1.0–2.05 ng/ml | 42 | 42 | 42 | 40 |
| *p,p'*-DDE quartile 4 | 2.05> ng/ml | 41 | 40 | 41 | 37 |
| HCB (median values) | | 0.19 ng/mL | 0.19 ng/mL | 0.190 ng/mL | 0.185 ng/mL |
| | | Number of boys | Number of boys | Number of boys | Number of boys |
| HCB quartile 1 | ≤0.15 ng/ml | 48 | 43 | 48 | 41 |
| HCB quartile 2 | 0.15–0.18 ng/ml | 29 | 37 | 29 | 35 |
| HCB quartile 3 | 0.18–0.26 ng/ml | 49 | 48 | 49 | 44 |
| HCB quartile 4 | 0.26> ng/ml | 39 | 37 | 39 | 35 |

The median maternal serum concentrations of PCB-153 was 0.45 ng/ml for the cases (range 0.03–1.7 ng/ml) and 0.47 ng/mL for the controls (0.03–4.0 ng/mL), of *p,p'*-DDE 1.1 ng/ml for cases (0.05–34 ng/ml) and 1.1 ng/mL for controls (0.05–20 ng/mL), and of HCB 0.19 ng/ml for cases (0.01–3.1 ng/ml) and 0.19 for controls (0.01–10 ng/ml). Levels of compounds in cases and controls, and numbers of men in the different exposure quartiles, are shown in Table 2.

When using the exposure variables as continuous variables, one ng/mL increase in PCB-153 levels corresponded to a 42% lower odds [OR of 0.58 (95%CI: 0.28–1.2)] of being a case (Table 3).

Correspondingly, one ng/mL increase in *p,p'*-DDE gave a 10% higher odds [OR of 1.1 (95% CI: 0.98–1.1)] of being a case, whereas one ng/mL increase in HCB gave an 8% lower odds [OR of 0.92 (95%CI: 0.63–1.3)] of being a case (Table 3).

When having the POP levels divided in quartiles, one unit increase in quartiles of PCB-153 corresponded to a 2% lower odds of being a case [OR 0.98 (95%CI: 0.76–1.3)]. For *p,p'*-DDE, one unit increase in quartiles corresponded to a 10% higher odds of being a case [OR 1.1 (95%

**Table 3. Odds ratios of a son with cryptorchidism depending on one unit increase in maternal levels (continuous or in quartiles) of POPs, unadjusted for levels of the other two POPs.**

| Compound | Exposure model | OR | 95%CI |
|---|---|---|---|
| PCB-153 | Continuous (ng/mL) | 0.58 | 0.28–1.2 |
| | In quartiles | 0.98 | 0.76–1.3 |
| *p,p'*-DDE | Continuous (ng/mL) | 1.1 | 0.98–1.1 |
| | In quartiles | 1.1 | 0.86–1.3 |
| HCB | Continuous (ng/mL) | 0.92 | 0.63–1.3 |
| | In quartiles | 1.0 | 0.81–1.2 |

CI: 0.86–1.3)], and for HCB one unit increase in quartiles corresponded to 0% higher odds [OR 1.0 (95%CI: 0.81–1.2)] (Table 3).

In the sensitivity analysis, only including boys born full term, all effect estimates remained similar (with the largest change being for PCB-153 as a continuous variable to an OR of 0.56, with the 95%CI still including 1.0). In our second sensitivity analysis, including all POPs as continuous variables simultaneously, one unit increase in the level of PCB-153 decreased the odds of being a case by 0.32% [OR 0.68 (95%CI: 0.32–1.5)], whereas one unit increase in the level of DDE increased the odds by 6.7% [OR 1.067 (95%CI: 0.99–1.2)] and one unit increase in HCB decreased the risk by 18% [OR 0.82 (95%CI: 0.46–1.5)].

## Discussion

We did not find any evidence of associations between maternal serum levels in early pregnancy of PCB-153, *p,p'*-DDE or HCB and the risk of cryptorchidism in the sons. This was robust to sensitivity analyses when only including boys born full term, and when adjusting for the levels of the corresponding two compounds.

A general strength of this study is that the samples analysed in this study were taken during the first trimester of pregnancy which largely includes the period when anti-androgenic exposures are suggested to disrupt the masculinisation of the human reproductive tract and the appropriate later development such as the testicular descent [21]. Levels of POPs in first trimester have also been reported to be correlated with levels in cord blood [12], which seems to strengthen the use of the available samples in this study as a suitable matrix.

There is a risk that some of the control boys might in fact have had cryptorchidism at an early age, since boys with a spontaneous ascent before the age of 6 months may not always be referred to a surgeon or urologist for a diagnosis ending up in the in-patient register [22]. This is illustrated by the lower reliability of registers than cohort study data on cryptorchidism [23]. This may, therefore, have led to an underestimation of the odds ratios for cryptorchidism of the different POPs in our study, especially since more subtle genital abnormalities (such as possibly spontaneously resolving cryptorchidism) more often may have environmental factors as components in the causation, as compared to more severe abnormalities [24]. There is also a risk that the measurement of the POPs to some extent has been misclassified [25], which we however may have partly mitigated by using levels not adjusted for serum lipids [26]. A potential risk of having misclassified the matching factors cannot be excluded but may be of a minor importance due to a fairly high quality of this type of data in the Swedish Medical Birth Register [27]. The possible effects on our results from residual confounding is hard to evaluate, since few firmly established environmental causes of cryptorchidism are known, except possibly maternal smoking [6] which we however matched for.

A strength of our study compared with other studies is that our study is about two times larger than another study reporting an association between a combination of the eight most abundant persistent pesticides, including DDE and HCB, in breast milk and cryptorchidism in sons [28]. We, further, measured the levels of exposure during the pregnancy which may be more relevant than measuring the exposure afterwards. Moreover, associations between a combination of compounds and cryptorchidism, such as in the study by Damgaard and coworkers [28], could be related to the metabolism of many different other exogenous and endogenous compounds, why studying one compound at a time as in our study may be advantageous. Our study is also about three times larger than a study reporting that the 2,3,7,8-TCDD equivalent quantity of several polychlorinated dibenzo-p-dioxins and furans, and dioxin-like PCBs, were associated with cryptorchidism, similar to their sum of PCBs which was close to statistically significant [29].

Our study did not include measurements of dioxins, furans, polybrominated diphenyl ethers, nor a sum of PCBs. Still, since PCB-153 has been reported to correlate well with total PCB levels [16, 17], we believe that our results would have been the same if we had measured total PCB exposure. Still, different PCB congeners have been reported to have different potential to affect sex hormone function such as PCB-138 but not PCB-153 being an antagonist to the androgen receptor function [30]–an effect that may be of specific relevance to cryptorchidism. Still these two compounds have been reported to be highly correlated [31] why PCB-153 levels also seems likely to indicate the levels of the anti-androgenic PCB-138. Nonetheless, since PCB-153 also has been reported to correlate with total dioxin toxic equivalents in human samples [32], this seems to indicate that also environmental exposure to overall dioxin-like toxicity in the maternal samples of our study was not associated with the risk of cryptorchidism in the boys. Another study on levels in breast milk reported that levels of PCB compounds seemed to protect against cryptorchidism, whereas levels of the specific compound octachlorodibenzofuran and polybrominated difenyl ethers were associated with a higher risk of cryptorchidism in a Danish cohort, albeit not in a Finnish cohort [33]. Without measurement of octachlorodibenzofuran and polybrominated difenyl ethers in our study, we cannot exclude that exposure to any of them could have been elevated in the mothers of our cases, and been a mechanism behind their cryptorchidism.

Nevertheless, although the levels of the measured compounds were lower in our study than in some other studies on exposure, such as for DDE [34–37], the lack of an association between exposure markers for levels of PCB, DDE and HCB and cryptorchidism in our study, seems to corroborate review articles indicating a general lack of a strong support for an association between exposure to potential endocrine-disrupting compounds and cryptorchidism as a specific outcome [6, 14], although DDE was reported associated with male reproductive disorders when taken together [14]. This may be of relevance to our study in which only DDE had a positive OR for the risk of cryptorchidism, remaining in the sensitivity analyses when all compounds were included simultaneously, albeit with a confidence interval including 1.0 in both cases.

Taken together, our study seems to corroborate previous studies indicating that prenatal exposure to PCB, DDT and HCB is not an important risk factor for the development of cryptorchidism.

## Author Contributions

**Conceptualization:** Anna Rignell-Hydbom.

**Formal analysis:** Jonatan Axelsson, Kristin Scott, Christian H. Lindh, He Zhang.

**Investigation:** Anna Rignell-Hydbom.

**Methodology:** Kristin Scott, He Zhang, Lars Rylander.

**Resources:** Joakim Dillner, Christian H. Lindh.

**Writing – review & editing:** Jonatan Axelsson, Kristin Scott, Joakim Dillner, Christian H. Lindh, He Zhang, Lars Rylander, Anna Rignell-Hydbom.

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
