## [Decision Letter · Decision Letter 0]

9 Jun 2020

PONE-D-20-11886

Exposure to polychlorinated compounds and cryptorchidism; a nested case-control study

PLOS ONE

Dear Dr. Axelsson,

Thank you for submitting your manuscript to PLOS ONE. After careful consideration, we feel that it has merit but does not fully meet PLOS ONE’s publication criteria as it currently stands. Therefore, we invite you to submit a revised version of the manuscript that addresses the points raised during the review process.

A marked-up copy of your manuscript that highlights changes made to the original version. You should upload this as a separate file labeled 'Revised Manuscript with Track Changes'.An unmarked version of your revised paper without tracked changes. You should upload this as a separate file labeled 'Manuscript'.

We look forward to receiving your revised manuscript.

Kind regards,

Angela Lupattelli, PhD

Academic Editor

PLOS ONE

Journal Requirements:

1. We noticed you have some minor occurrence of overlapping text with the following previous publication(s), which needs to be addressed:

https://journals.plos.org/plosone/article?id=10.1371%2Fjournal.pone.0044767

In your revision ensure you cite all your sources (including your own works), and quote or rephrase any duplicated text outside the methods section. Further consideration is dependent on these concerns being addressed.

Additional Editor Comments:

Dear authors,

- please avoid using the terminology "statistically significant" but rather evaluate the findings of your work on the basis of the 95% CI of the observed effect estimates.

- it would be informative to have a table summarizing the association meausures identified in the study.

- the discussion needs some further elaboration as to why the findings do/do not align with prior research, and also in light of the comments raised by the reviewers concerning risk of misclassification of exposure, residual confounding and unmeasured confounding.

Reviewers' comments:

Reviewer's Responses to Questions

**Comments to the Author**

1. Is the manuscript technically sound, and do the data support the conclusions?

Reviewer #1: Yes

Reviewer #2: Partly

2. Has the statistical analysis been performed appropriately and rigorously? 

Reviewer #1: Yes

Reviewer #2: Yes

3. Have the authors made all data underlying the findings in their manuscript fully available?

Reviewer #1: No

Reviewer #2: Yes

4. Is the manuscript presented in an intelligible fashion and written in standard English?

Reviewer #1: Yes

Reviewer #2: Yes

5. Review Comments to the Author

Reviewer #1: The authors presented a study on the association between polychlorinated compounds and cryptorchidism using linkage of a Swedish biobank and national health registers. Although the research question is not new it may be relevant to study this in a relative large sample size of 165 cases and 165 controls.

My major comments to the manuscript includes

1) I would prefer to have a descriptive table on the characteristics mathced for - and although these by design should be equal in the case and control group it would be nice to see how well they fit.

2) On line 129-131 the authors stated that they decided not to include other exposures due to high correlations. The correlations reported were moderate and I would like to see the results of including additional exposures as a subanalysis or some kind of combined measure to compare with what was reported in previous studies.

3) I would prefer to see the results presented in a table in addition to the text.

4) P values from the results should be deleted since CIs gives the necessary information and p-values may be misleading since they are highly sample size dependent.

5) In the discussion add limitations: discussion of potential misclassification of exposure, outcome and matching factors and potential residual confounding.

Reviewer #2: This study investigated the associations between cryptorchidism and POPs measured in first trimester maternal serum. The results showed no significant associations. However, the authors only measured only three chemicals as representatives for POP exposures.

1. Line 70. Why do the authors collect first trimester maternal serum?

2. Line 72. Although the authors believe that PCB-153 can be a representative for PCB exposures. The individual PCBs may produce effects simultaneously.

3. Line 116. Why does this paragraph only contain one sentence?

6. PLOS authors have the option to publish the peer review history of their article (what does this mean?). If published, this will include your full peer review and any attached files.

Reviewer #1: No

Reviewer #2: No

---

## [Author Response · Author response to Decision Letter 0]

3 Jul 2020

Response to reviewers regarding PLOS ONE Decision: Revision required [PONE-D-20-11886] - [EMID:0c61a118f2ed1618] 

Dear Editor,

We want to thank you and the reviewers for the wise comments which here will be addressed.

Our answers are shown in italics below each comment.

With best regards on behalf of all co-authors,

Jonatan Axelsson, MD, PhD

Lund University

Sweden

1. We noticed you have some minor occurrence of overlapping text with the following previous publication(s), which needs to be addressed:

https://journals.plos.org/plosone/article?id=10.1371%2Fjournal.pone.0044767

In your revision ensure you cite all your sources (including your own works), and quote or rephrase any duplicated text outside the methods section. Further consideration is dependent on these concerns being addressed.

-The text has been gone through, but the mentioned overlappings have not been clearly identified. Still, to highlight the similarity to the mentioned publication, a reference to the publication has been done on page 4, line 71-72.

Additional Editor Comments:

Dear authors,

- please avoid using the terminology "statistically significant" but rather evaluate the findings of your work on the basis of the 95% CI of the observed effect estimates.

-We have now deleted the part of the sentences regarding statistical significance.

- it would be informative to have a table summarizing the association meausures identified in the study.

-Such a table (Table 3) has now been added, and can be found on page 11.

- the discussion needs some further elaboration as to why the findings do/do not align with prior research, and also in light of the comments raised by the reviewers concerning risk of misclassification of exposure, residual confounding and unmeasured confounding.

-We agree that the discussion could benefit from being more precise, and now changed it to mention the review articles on the topic directly, and removed the references to original articles (page 15). We, thereafter, added the fact that DDE was associated with male reproductive disorders as a common entity in one of the review articles, and that this compound was the only one with a positive OR for the risk of cryptorchidism in our study. 

5. Review Comments to the Author

Reviewer #1: 

My major comments to the manuscript includes

1) I would prefer to have a descriptive table on the characteristics mathced for - and although these by design should be equal in the case and control group it would be nice to see how well they fit.

-Such a table has now been added as Table 1, page 8

2) On line 129-131 the authors stated that they decided not to include other exposures due to high correlations. The correlations reported were moderate and I would like to see the results of including additional exposures as a subanalysis or some kind of combined measure to compare with what was reported in previous studies.

-We have now added an analysis with a simultaneous inclusion of the three different exposure markers. This is mentioned in the end Methods section as a second sensitivity analysis, page 8. The results of this analysis is written on page 12 in the end of the Results section of the manuscript, as well as in the first paragraph of the Discussion.

3) I would prefer to see the results presented in a table in addition to the text.

-Such a table (Table 3) of the main analysis has now been added, and can be found on page 11.

4) P values from the results should be deleted since CIs gives the necessary information and p-values may be misleading since they are highly sample size dependent.

-The p values have now been deleted from the manuscript.

5) In the discussion add limitations: discussion of potential misclassification of exposure, outcome and matching factors and potential residual confounding.

-We have now added a discussion about a possible misclassification of the outcome (page 13). After this discussion we changed the formulation of the beginning of the next paragraph (line 213), and also added an additional reference to the mentioning of the study by Damgaard et al for clarification (reference 28). We have thereafter also added a discussion of possible misclassification of the exposure, and of the matching factors. It is hard to evaluate the risk of possible residual confounding since few environmental risk factors, are known. This has also been added to the discussion. 

Reviewer #2: This study investigated the associations between cryptorchidism and POPs measured in first trimester maternal serum. The results showed no significant associations. However, the authors only measured only three chemicals as representatives for POP exposures.

-We have now added some reasoning around the possible relevance also for other compounds not measured (page 14, line 228-).

1. Line 70. Why do the authors collect first trimester maternal serum?

- The samples available from the biobank were taken in early pregnancy when the screening for rubella is done in Sweden. Still, we believe that these levels are representative due to correlations reported between POP levels in first trimester and levels in cord blood. This is now mentioned in the Discussion, page 13 (line 195-197).

2. Line 72. Although the authors believe that PCB-153 can be a representative for PCB exposures. The individual PCBs may produce effects simultaneously.

-It is true that individual PCBs may produce effects simultaneously. Still, at least two studies reported that PCB-153 was a good marker for the sum of PCBs in serum. We now added one of these references to the discussion on page 14, line 227 (reference 17). We have also added the correlation between PCB-153 and the possibly antiandrogenic PCB-138 on line 228-232.

3. Line 116. Why does this paragraph only contain one sentence?

-This sentence was separated from the rest of the paragraph to make it more easily discernible, but has now been moved as a final sentence to the paragraph above, since it is still related to the determination of the compounds.

As an additional change during the revision process we have limited the number of decimals in the ranges of the levels of the POPs in the different quartiles in Table 2.

---

## [Editor Report · Decision Letter 1]

8 Jul 2020

Exposure to polychlorinated compounds and cryptorchidism; a nested case-control study

PONE-D-20-11886R1

Dear Dr. Axelsson,

We’re pleased to inform you that your manuscript has been judged scientifically suitable for publication and will be formally accepted for publication once it meets all outstanding technical requirements.

Kind regards,

Angela Lupattelli, PhD

Academic Editor

PLOS ONE
---

## [Editor Report · Acceptance letter]

15 Jul 2020

PONE-D-20-11886R1 

Exposure to polychlorinated compounds and cryptorchidism; a nested case-control study 

Dear Dr. Axelsson:

I'm pleased to inform you that your manuscript has been deemed suitable for publication in PLOS ONE. Congratulations! Your manuscript is now with our production department. 

Kind regards, 

on behalf of

Dr. Angela Lupattelli 

Academic Editor

PLOS ONE